

# Observationally constrained analysis of sea salt aerosol in the marine atmosphere

Huisheng Bian[1,2], Karl Froyd[3,4], Daniel M. Murphy[3], Jack Dibb[5], Mian Chin[2], Peter R. Colarco[2],

Anton Darmenov[2], Arlindo da Silva[2], Tom L. Kucsera[6], Gregory Schill[3,4], Hongbin Yu[2], Paul

Bui[7], Maximilian Dollner[8], Bernadett Weinzierl[8], and Alexander Smirnov[9]

[1] University of Maryland at Baltimore County, Baltimore County, MD

[2] NASA Goddard Space Flight Center, Greenbelt, MD

[3] NOAA Earth System Research Laboratory, Chemical Sciences Division, CO

[4] Cooperative Institute for Research in Environmental Sciences, University of Colorado, Boulder, CO

[5] University of New Hampshire, Durham, NH

[6] Universities Space Research Association, Columbia, MD

7 NASA Ames Research Center, Moffett Field, CA

8 University of Vienna, Faculty of Physics, Aerosol and Environmental Physics , Boltzmanngasse 5, A-
1090 Wien, Austria

9 Science Systems and Applications, Inc., Lanham, MD 20706

**Abstract**

Atmospheric sea salt plays important roles in marine cloud formation and atmospheric

chemistry. We performed an integrated analysis of NASA GEOS model simulations run

with the GOCART aerosol module, in situ measurements from the PALMS and SAGA

instruments obtained during the NASA ATom campaign, and aerosol optical depth

(AOD) measurements from AERONET Marine Aerosol Network (MAN) sun

photometers and from MODIS satellite observations to better constrain sea salt in the

marine atmosphere. ATom measurements and GEOS model simulation both show that



sea salt concentrations over the Pacific and Atlantic oceans have a strong vertical
gradient, varying up to four orders of magnitude from the marine boundary layer to free
troposphere. The modeled residence times suggest that the lifetime of sea salt particles
with dry diameter less than 3 μm is largely controlled by wet removal, followed next by
turbulent process. During both boreal summer and winter, the GEOS simulated sea salt
mass mixing ratios agree with SAGA measurements in the marine boundary layer (MBL)
and with PALMS measurements above the MBL. However, comparison of AOD from
GEOS with AERONET/MAN and MODIS aerosol retrievals indicated that the model
underestimated AOD over the oceans where sea salt dominates. The apparent discrepancy
of slightly overpredicted concentration and large underpredicted AOD could not be
explained by biases in the model RH, which was found to be comparable to or larger than
the in-situ measurements. This conundrum is at least partially explained by the sea salt
size distribution; where the GEOS simulation has much less sea salt percentage-wise in
the smaller particles than was observed by PALMS. Model sensitivity experiments
indicated that the simulated sea salt is better correlated with measurements when the sea
salt emission is calculated based on the friction velocity and with consideration of sea
surface temperature dependence than that parameterized with the 10-m winds.


**1.  Introduction**
Bubble bursting and jet drops at the ocean surface result in the production of sea spray
particles composed of inorganic sea salt and organic matter (e.g., Quinn and Bates, 2013).
Among various atmospheric aerosol components, sea salt is estimated to have the largest



mass emission flux and the second largest atmospheric mass loading globally (Textor et
al., 2006). Sea salt particles in the atmosphere could exert direct radiative effect of
around -1.5 to -5.03 W/m$^2$ annually at the top of atmosphere (IPCC, 2001). On a global
and annual scale, the direct radiative effect of sea salt is equal to or greater in magnitude
than that of natural sulfate and soil dust (Jacobson, 2001; Takemura et al., 2002). Sea salt
particles are efficient cloud condensation nuclei (CCN). Consequently, sea salt particles
have indirect effects on climate and weather (Dadashazaer et al., 2017; Dall et al., 2017;
Kogan et al., 2012; Pierce and Adams, 2006; Quinn et al., 2017).  Furthermore, sea salt
aerosol particles serve as sinks for reactive gases and small particles and are a source of
halogens to the atmosphere (e.g., Alexander al., 2005; Anastasio et al., 2007; Lawlet et
al., 2011). There is also observational evidence suggesting that new particle formation
may be suppressed in the presence of sea salt aerosol (Browse et al., 2014; Lewis and
Schwartz, 2004). To quantify the effects of sea salt aerosol on the environment, a detailed
knowledge of its mass, size, and vertical distribution is required.

We present a comprehensive evaluation of sea salt aerosol simulated with the Goddard
Chemistry, Aerosol, Radiation, and Transport model (GOCART) in the Goddard Earth
Observing System (GEOS) framework using aerosol measurements obtained during the
Atmospheric Tomography Mission (ATom). ATom is a NASA-funded Earth Venture-
suborbital project. ATom deployed an extensive gas and aerosol instrumental payload on
the NASA DC-8 aircraft for systematic, global-scale sampling of the atmosphere,
profiling continuously from 0.2 to 12 km altitude with flight routes over the Pacific,
Atlantic, Southern Ocean, North America and Greenland from 85°N to 65°S (see Fig. 1).



Flights occurred in four seasons over a 3-year period (2016-2018) and we study the first
two ATom measurements that represent the summer and winter seasons for both
hemispheres.  The ATom data provides an unprecedented opportunity for models to
evaluate transport and parameterizations of physical and chemical processes. This work
utilizes ATom's high frequency vertical measurements of sea salt over global remote
oceans from marine boundary layer (MBL) to the upper troposphere, in contrast with
typical model validation of sea salt simulation with in situ measurements at ground
surface limited to selected locations and regions (Kishcha et al., 2011; Spada et al., 2013,
2015; Tsyro et al., 2011; Witek et al., 2007), and commonly using monthly averaged
observations (Grini et al., 2002;  Textor et al., 2006).

In this study, we examine sea salt in MBL using both the ATom measurements and
GEOS GOCART simulations. We explore sea salt vertical distribution in various
latitudinal zones over the Pacific and Atlantic oceans to investigate simulated dry and wet
deposition processes. Finally, we examine the sea salt size distribution, important to both
AOD calculations and cloud formation.

The GEOS/GOCART model is described in section 2, particularly the different sea salt
emission schemes tested in this study. The NASA ATom field campaign is introduced in
section 3, including a brief description of the Particle Analysis by Laser Mass
Spectrometry (PALMS) and Soluble Acidic Gases and Aerosols (SAGA) instruments that
are used to provide sea salt measurements. The measurements and model results are
presented in section 4 and the emission, removal processes, vertical profile, size



distributions, and AOD are analyzed. In section 5, we summarize our sea salt study and
discuss the potential important chemical/physical processes that could have an impact on
sea salt simulation for future improvement.

**2.  Model description**
Global sea salt is simulated by using GEOS/GOCART, which is a global aerosol model
GOCART (Chin et al., 2002, 2009, 2014) implemented in the GEOS Earth system model
(Gelaro et al., 2017; Rienecker et al., 2011). The GEOS/GOCART aerosols include dust,
sea salt, sulfate, nitrate, ammonium, black carbon, and organic matter, mixed externally
(Bian et al., 2013; 2017; Colarco et al., 2010).

The sea salt emission scheme in the GEOS/GOCART model was initially based on the
algorithm of Gong (2003) who provided a parameterization of the size-resolved flux of
sea salt particles as a function of the 10-m wind speed. Two modifications to this scheme
were subsequently developed based on comparisons of simulated sea salt aerosol to
satellite AOD from the Moderate Resolution Imaging Spectroradiometer (MODIS)
(Darmenov et al., 2013; Randles et al., 2017): 1) the emission function was recalibrated
in terms of the surface friction velocity rather than the 10-m wind speed and 2) a sea
surface temperature (SST) correction term was introduced. This new emission algorithm
with both surface wind and temperature modifications is used in the main body of the
paper and a detailed description of the emission is given in supplementary material. We
examined the three sea salt emission schemes using ATom measurements and the results
are given in supplementary material.




The current default setting of GEOS/GOCART allows sea salt to be completely removed
by warm clouds from convective updraft and from large-scale rainout and washout. Sea
salt can also be removed by dry deposition (turbulent) and sedimentation. These
processes were described in Chin et al. (2002). We assume that the particles undergo
hygroscopic growth according to the equilibrium parameterization of Gerber (1985),
which is a function of the relative humidity (RH). The humidified particle sizes are
considered in our computations of the particle sedimentation, aerodynamic deposition
velocity, and optical properties.

The GEOS/GOCART includes five bulk sea salt size bins in the range of 0.06-20 μm in
dry diameter. Specifically, they are 0.06-0.2, 0.2-1.0, 1.0-3.0, 3.0-10, and 10-20 μm,
respectively. The first bin was added to facilitate aerosol-cloud and optical property
studies (Colarco et al. 2010), which was not included in the previous GOCART versions
(Chin et al., 2002, 2014). The sea salt particle density is 2200 (kg/m$^3$) for all sizes.

In this study, we ran GEOS/GOCART at a global ~50 km horizontal resolution on the
cubed-sphere grid and 72 vertical layers from surface up to 0.01mb. We ran the model in
"replay" mode, which sets the model dynamical state (winds, pressure, and temperature)
at every 6 hours to the balanced state provided by the meteorological reanalysis fields
from the Modern-Era Reanalysis for Research and Applications version 2 (MERRA-2).
One and half year simulation was conducted from the beginning of 2016 to cover the first



two phases of ATom measurement periods, with the first half year of the simulation used
as a spin up period.

**3.  ATom aircraft sea salt measurement from PALMS and SAGA**
ATom provides measurements for various important atmospheric gases, aerosols and
their precursors over vast open oceans. Among these, sea salt has been measured by two
instruments, the NOAA PALMS instrument, which provides mass mixing ratio and size
distribution up to 3 μm in dry diameter, and the University of New Hampshire SAGA
instrument, which includes measurements of sodium ion, a good sea salt proxy.
PALMS is a laser ionization mass spectrometer which makes in situ measurements of the
chemical composition of individual aerosol particles. A detailed description of PALMS,
including its physical working mechanism and measurement features, has been given by
Murphy et al., (2018). The instrument is capable of measuring particles from 0.12 to 3
μm in dry diameter and analysis is completed in less than 1 millisecond after the aerosols
enter the inlet. The real power of the PALMS sea salt measurements is twofold: a) high
sensitivity at low concentrations above the MBL such that the measured vertical profiles
are more reliable than most previous data, and b) the data are size-segregated up to 3 μm
in dry diameter, covering the active size range for optical and radiation calculations.
In the cloud-free MBL, sea salt concentrations inferred from the SAGA sodium data are
highly correlated with PALMS sea salt measurements (Murphy et al., 2018). SAGA
measures sodium ions extracted from the aerosol. A factor of 3.27 is applied to convert
the SAGA measured sodium mass to total sea salt mass (Keene et al., 1986; Wilson,





1975). This assumes that all of the measured sodium comes from sea salt, which should
be a reasonable assumption for most ATom samples. SAGA collects particles on a filter
with a sampling frequency of around 5-15 minutes to allow more time for the filter media
to collect sufficient particles. As reported by the DC-8 Inlet Characterization Experiment
(DICE), the SAGA inlet performed nearly identically in the marine boundary
environment to the U. Hawaii inlet used by PALMS during ATom (McNaughton et al.,

170    2007).


We use ATom1 (Jul.-Aug., 2016) and ATom2 (Jan.-Feb., 2017) campaign data in this
study. These two deployments combined together provided detail information of summer
and winter on a global scale.

**4.  Results and Discussions**

**4.1 Comparison in marine boundary layer**
Sea salt is sufficiently rich in the MBL that SAGA can collect enough aerosol there for
analysis. Comparisons of the sea salt in a layer from surface up to 1.5 km between the
model simulation and ATom (PALMS and SAGA) measurements are shown in Fig. 2a.
To have a reasonable comparison, we conducted three data treatments. First, we excluded
SAGA samples with significant dust signal when the measurements meet the two
conditions: sample $Ca^{2+}$ larger than 0.05 µg/sm$^3$ and the ratio of $Ca^{2+}$ to $Na^+$ larger than
0.06. Second, we only include GEOS sea salt particles smaller than 3 µm in dry diameter
to be consistent with the instrument measurements. Third, we sampled GEOS and



PALMS data at the SAGA measurement time frequency when the SAGA has valid
measurements. The agreement between model and measurement is good. The correlation
coefficients are generally higher than 0.79 for both GEOS-PALMS and GEOS-SAGA in
both ATom1 and 2 periods.

There are outliers on the Figure 2a. Just a small amount of cloud can wash off salt
previously deposited on an inlet wall. Therefore, in Figure 2b we excluded samples that
could be contaminated by clouds during sampling, using cloud indicator data from the
Cloud, Aerosol, and Precipitation Spectrometer (CAPS). The outliers are gone on Figure
2b and the correlation coefficients between model and measurements are indeed
improved a little bit, i.e. larger than 0.85. On the other hand, the GEOS sea salt mass
mixing ratios are still more than double of those of PALMS (2.3 in ATom1 and 4.7 in
ATom2), which could be at least partially explained by potential sampling biases in
PALMS instrument, particularly in the size distribution. The cutpoint of 3 μm in dry
diameter recommended by instrument teams used in this study is subject to a large
uncertainty of wet/dry size ratio that is strongly dependent on ambient relative humidity.
Furthermore, the sea salt mass distribution is (sometimes) still rising sharply through the
inlet cutpoints. Considering the combination of all these systematic and random
uncertainties, which are decreased across the sea salt coarse mode, the measurement can
easily result in uncertainties on the order of ~x2 in dry mass. When checking the
comparison between GEOS and SAGA, GEOS sea salt mixing ratio is comparable to or
slightly larger than SAGA results (i.e. 0.92 in ATom1 and 1.3 in ATom2). Overall, the
GEOS sea salt is most likely to overestimate sea salt mass during southern hemisphere





summer period. Comparing sea salt between the two instruments directly shows a high
correlation (0.76 in ATom1 and 0.90 in ATom2) as well (also see Murphy et al., 2018).

**4.2 Vertical distribution**
Understanding the sea salt vertical distribution is important for climate studies,
particularly in the tropical marine upper troposphere where a reliable background aerosol
field is needed. However, most previous sea salt measurements were limited to the
surface or near coastal areas, leading to nearly no in situ observations of the vertical
distribution of sea salt over vast areas of the open oceans. The ATom measurements fill
this gap by providing atmospheric tomography measurements over the Pacific, Atlantic,
and Southern oceans from near surface to the upper troposphere (0.2-12 km).
Furthermore, the PALMS instrument measures in situ sea salt mass and size distribution.
A good sensitivity of the PALMS measurements makes it very useful in studying the
relatively clean environments above the MBL. Using the ATom sea salt measurements
over remote open oceans has some additional advantages versus previous studies. For
instance, airborne measurements alleviate biases typical at land stations due to onshore
wave breaking activities, especially at sites with steep topography (Witek et al., 2007;
Spada et al., 2015).

Figure 3 shows the sea salt vertical profiles of PALMS measurement and GEOS model
simulation over 5 latitudinal zones over Pacific and Atlantic oceans in ATom1 and
ATom2. The GEOS model results are sampled at the time and location closest to the
measurement points. As discussed in section 4.1, modeled sea salt mass concentrations





are higher than the PALMS data near the surface over all latitudinal zones during both
summer and winter seasons. There are often two regimes vertically with a sharp gradient
in the lower atmosphere and a lesser gradient above. Wet removal processes, particularly
convective cloud removal, are likely the driving factors for the sea salt distribution in the
size range considered in this study (Table 1a). Sea salt is a highly soluble species. It is
assumed to fully dissolve into clouds, which results in efficient removal by shallow
marine clouds, typically marine stratus and stratocumulus clouds (Eastman et al., 2011,
Lebsock et al., 2011, Wood 2012, Zhou et al., 2015). Sea salt dry deposition (turbulent)
and sedimentation also contribute to its removal in low altitudes. Interestingly,
sedimentation process plays the smallest removal role for the sea salt particles studied in
this work, while it overwhelmingly controls sea salt loss rate (i.e. more than 1.5 times
those of all other processes combined) when coarser mode sea salt is included, see Table
1b. This is understandable because nearly 90% of injected sea salt particles are in coarse
mode. Since sea salt is found mostly in the lower atmosphere, further removal of sea salt
particles by cold clouds was found to have only marginal impact on its mass budget in
our sensitivity studies, although its feedback on cold clouds needs further studies. Note
that results in Table 1a and b are summarized on an annual basis from July 2016 to June

250    2017.


Atmospheric convection impacts the sea salt vertical distribution as well. The height of
the turnaround level (or the transition layer) between two vertical distribution regimes in
Fig. 3 is around 600 hPa in the polar regions and moves up to 400 hPa in the tropical
region, given that more vigorous convective activities occur in the tropical region. The



seasonal variation of the vertical gradient is larger in polar regions than in tropical region,
consistent with stronger seasonal variations of the meteorological fields (e.g. T, RH,
wind, etc) in high latitudes.

**4.3 Marine aerosol AOD**
To provide an overall picture of sea salt for this study, we compared the GEOS AOD
with satellite MODIS Collection 6 (C6) Aerosol AOD retrieval (Levy et al., 2013) and
AERONET Maritime Aerosol Network (MAN) measurements (Smirnov et al., 2017) by
focusing on sea salt dominant regions. AOD integrates extinction by all aerosol in the
atmospheric column, with extinction dependent on the absolute mass, size distribution,
hygroscopic growth, vertical distribution, and optical property of each individual
component and the composition of aerosols.

Figure 4 shows total AOD comparison between MODIS and GEOS in August 2016 and
February 2017. Here, the GEOS AODs are sampled using daily MODIS AOD retrieval.
The AODs are only shown where the fraction of sea salt mass relative to the total aerosol
mass simulated by GEOS (bottom panel) is larger than 0.7 so that we can focus our
discussion over sea salt dominant regions. GEOS AODs are much lower than MODIS
AODs for both seasons over remote oceans where sea salt dominates. Even after
improvements, MODIS C6 AOD remains a positive bias up to 0.03 at low AOD (Figure
16 in Levy et al., 2013). It is difficult for us to remove this bias in the comparison shown
in the Figure 4. Another AOD comparison between AERONET MAN and GEOS,





therefore, is explored since there is no positive systematic bias reported in MAN's
measurement.

The conclusion of a lower GEOS AOD can also be obtained in Fig. 5 by comparing AOD
between the MAN cruise measurement and the GEOS simulation that occurred from July,
2016 to June 2017. AERONET MAN provides ship-borne aerosol optical depth
measurements from the Microtops II sun photometers. The GEOS model results are
sampled at the time and location of the ship-based measurements. The model AODs are
much smaller than MAN measurements over a majority of the open ocean areas except
part of the Atlantic Ocean where AOD was impacted by the dust. The scatter plot at the
bottom of the figure indicates clearly that the modeled AOD is biased low, especially
over the Southern Ocean where the model AOD is less than half of MAN's.

On the one hand, GEOS' sea salt mass is comparable to SAGA *in situ* measurements in
the MBL, and on the other hand, GEOS underestimates AOD when compared with
measurements from MAN and MODIS. The agreement with PALMS vertical gradients
shows that the AOD cannot be explained by sea salt above the MBL. There are various
potential reasons for this conundrum, such as the sea salt size distribution, atmospheric
relative humidity, sea salt particle hygroscopic growth rate, sea salt refractive index, etc.
We will discuss the first two potential reasons below.

**4.4 Size distribution and atmospheric RH**





The sea salt size distribution is a key factor in AOD calculation because small particles
are more optically efficient. Sea salt size distribution also affects AOD calculation by
affecting sea salt mass distribution via sea salt transport and removal processes. The
necessity to study sea salt size distribution lies also in that it plays an important role in
atmospheric chemistry, radiative effects, and cloud formation processes.

To quantify size impact, we calculate normalized percentage of sea salt mass in each of
the first three size bins for PALMS and GEOS over three atmospheric vertical layers for
ATom1 and 2, as shown in Figure 6. The three vertical layers (i.e. 0-1.5, 1.5-6, and >6
km) represent the boundary layer, middle troposphere, and upper troposphere. GEOS sea
salt particle mass and size have been computed at RH of 45% to match the measurement
condition of PALMS. Although the particle sizes are limited to be less than 3 μm in dry
diameter here due to PALMS measurements, we are more interested in the small particles
since they are optically important and are more important in cloud formation on a per unit
mass basis. The amplitude of the distribution is much shallower in PALMS than in
GEOS. In other words, with the same sea salt mass, the fraction of sea salt in the finest
mode in PALMS would be much more (i.e. about 5-7 times higher) than in GEOS. To
quantify the potential impact of sea salt size distribution on AOD calculation, we
calculate the sea salt mass extinction efficient (MEE) integrated over the three bins using
the two size distributions of PALMS and GEOS at RH 45% and 550 nm in the same three
vertical layers and in the whole atmosphere (Table 2). The size segregated MEEs used in
the calculation are 1610.3618, 5622.7075, and 1216.4149 $m^2$ $kg^{-1}$ for the bins 1-3,
respectively. The integrated MEE of GEOS (1679.36 $m^2$ $kg^{-1}$) is only 76.2% of that of



PALMS (2203.67 $m^2$ $kg^{-1}$). Thus, the underestimation of GEOS AOD shown in Figure 5c
is partially stemmed from the sea salt size distribution. The underestimation of AOD by
GEOS is more significant in low atmosphere shown in Table 2, which implies that the sea
salt size distribution from emission may need to be revisited.

Apparently, sea salt size distribution is a potential culprit for the dichotomy in GEOS
simulation since GEOS partitions more sea salt onto larger particles which are less
optically active compared with the significant fine sea salt mode observed in PALMS
measurements. Such large underestimation of fine sea salt particles by the model may
have significant implications not only on the AOD calculation but also on studies of
radiative effects and cloud formation because particle number concentration is a key
quantity for these processes. The conclusion that GEOS sea salt size distribution favors
the coarse mode sea salt particles is consistent with a recent study of Naumann et al.,
(2016), which found that the sea salt emission of Gong (2003) yielded overestimations in
the PM10 measured at coastal stations and underestimations at inland stations over
northwestern Europe.

Sea salt particle size distribution changes horizontally and vertically, but the change is
much smaller than the difference between those of model and measurement. This implies
a possibility of using a global size distribution without sacrificing much accuracy.

Atmospheric water, another possible reason for the AOD underestimation, was also
investigated. Figure 7 compares atmospheric RHs between ATom measurements and



GEOS simulations along flight tracks summarized over the same regions as in Fig. 3.
Almost everywhere the model's RH is higher than ATom measurement, including MBL
where humidity is typically high, with only a few exceptions. Thus, atmospheric water
simulation is not responsible for the low AOD calculation. In fact, using measured RH
along with the model's sea salt size distribution and vertical distribution would give even
lower AOD. There should be other factors contributing to a lower GEOS AOD
calculation as well, such as sea salt hygroscopic growth rate, sea salt optical properties,
and other aerosol species over ocean. Further investigations for these factors are needed
to better understand the GEOS sea salt simulation.

**5.  Conclusions**
A systematic and comprehensive global sea salt study was conducted by integrating
NASA GEOS model simulations with ATom in situ measurements from the PALMS and
SAGA instruments, as well as AOD measurements from AERONET MAN and satellite
MODIS over the oceans. This work takes advantage of PALMS sea salt vertical profile
measurement together with SAGA filter measurements in MBL. The study covers global
remote regions over the Pacific, Atlantic, and Southern Oceans from near the surface to
~12 km altitude and covers both summer and winter seasons. Important atmospheric sea
salt fields, e.g. mass mixing ratio, vertical distribution, size distribution, and aerosol
AOD, are examined. The meteorological field of RH and the sea salt simulation
processes of emission, dry deposition, sedimentation, and large scale and convective wet
depositions were explored to explain the sea salt fields and to reveal a potential direction
for model improvement.






Generally, the agreement between ATom measurements and the model is remarkable,
both in terms of absolute loading and especially in the shape of the vertical distribution
under a huge variety of tropospheric environments. The correlation coefficients are
generally higher than 0.8 between GEOS-PALMS and GEOS-SAGA for both ATom1
and ATom2 periods. GEOS results captured the strong sea salt vertical gradient shown in
the measurements except over SH high latitudes, where the PALMS's gradient is deeper.
In the MBL, the current GEOS sea salt simulation is comparable (ATom1) or slightly
higher (ATom2) than SAGA data, which in turn is higher than PALMS data.

An underestimation of GEOS aerosol AOD over sea salt dominated oceans was found
from the comparison of AODs between GEOS and MAN, as well as GEOS and MODIS.
This is contradictory to the finding that GEOS sea salt mass abundance is comparable to
or slightly higher than measurements. This conundrum may be partially resolved by the
sea salt mass size distributions compared between GEOS and PALMS. The GEOS sea
salt mass size distribution favors the coarse mode while PALMS has a larger fraction of
more optically active submicron sea salt. The atmospheric water field, however, can be
ruled out as the cause of model underestimation of AOD, since the GEOS RH is
comparable to or higher than ATom measurements almost everywhere along the flight
tracks, especially in MBL.

Atmospheric sea salt vertical distribution is impacted by various processes including
emission, hygroscopic growth, dry deposition, sedimentation, wet deposition, convection,

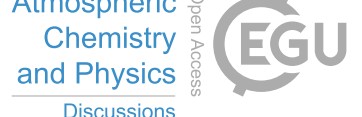

and large-scale advection. Among these processes, wet deposition, owing to both shallow
marine cloud structure and rapid hygroscopic growth of sea salt particles, is most
important in shaping the vertical profile for the size range studied in this work and results
in a sharp gradient in the low atmosphere where RH is typically very high. Vertical
convection is also important.

More work is needed in the future to investigate sea salt hygroscopic growth rate, optical
properties, sea water salinity, sea ice, and marine organic aerosol to understand the
dilemma in GEOS simulation. Sea water salinity, which is missing in the GEOS aerosol
model, has an impact not only on sea spray emission but also on sea spray aerosol (SSA)
size. Barthel et al. (2014) reported that the dry SSA size distribution shifts towards
smaller sizes with lower salinities found in the EMEP intensive campaigns. Sea ice,
whose contribution is also missed in the GEOS aerosol model, could be an important
source of sea salt aerosol over polar regions and has significant implications for polar
climate and atmospheric chemistry reported by recent publications (Dall et al., 2017; May
et al., 2016; Rhodes et al., 2017). More importantly, primary marine organic aerosols
(Randles et al., 2004), which come also from sea spray bubble bursting as sea salts but
are more submicron particles, should be investigated to disentangle the sea spray
aerosols.

**Author contribution**
Huisheng Bian and Mian Chin designed the experiments. Peter R. Colarco, Anton Darmenov,
Arlindo da Silva, Tom L. Kucsera, and Hongbin Yu contributed to GEOS-GOCART model setup
and provided tools to analyze model data. Huisheng Bian conducted the model simulation and in



charge of the analyses. Karl Froyd, Daniel M. Murphy, and Gregory Schill provided ATom
PALMS measurement data. Jack Dibb provided ATom SAGA measurement data. Maximilian
Dollner and Bernadett Weinzierl provided ATom CAPS cloud data. Paul Bui provided ATom
MMS data for RH measurement. Hongbin Yu and Alexander Smirnov provided MODIS satellite
and AERONET MAN measurement data. All authors contributed to the data analyses and paper
writing.


**Acknowledgments**
This research was support by two programs of the National Aeronautics and Space
Administration (NASA): Atmospheric Composition: Modeling and Analysis Program
(ACMAP) and Earth Venture-suborbital program for the Atmospheric Tomography
Mission (ATom).

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





**Table 1a**. GEOS sea salt budget analysis for the particles up to 3 μm in dry diameters
using the three emission algorithms on annual basis from July 2016 to June 2017

|  | Emi1 | Emi2 | Emi3 |  |
|---|---|---|---|---|
| Emission (Tg/yr) | 408.8 | 615.6 | 515.2 |  |
| Burden (Tg) | 1.21 | 1.88 | 1.63 |  |
| Lifetime (days) | 1.08 | 1.12 | 1.16 |  |
| Surf concentration (μg/kg) | 2.5 | 3.9 | 3.2 |  |
| Dry deposition (Tg/yr) | 82.4 | 127.0 | 103.1 |  |
| Sedimentation (Tg/yr) | 60.7 | 88.7 | 61.1 |  |
| Kdry (days$^{-1}$) | 1.37 | 1.32 | 1.17 |  |
| Wet deposition (Tg/yr) | 123.8 | 181.9 | 140.3 |  |
| SV deposition (Tg/yr) | 142.2 | 218.8 | 211.8 |  |
| Kwet (days$^{-1}$) | 0.45 | 0.43 | 0.44 |  |
|  |  |  |  |  |

**Table 1b**. Similar to Table 1a but for all particle size range

|  | Emi1 | Emi2 | Emi3 | AeroCom |
|---|---|---|---|---|
| Emission (Tg/yr) | 3185.7 | 4797.6 | 4015.5 | 2190-117949 |
| Burden (Tg) | 4.79 | 7.55 | 6.80 | 3.4-18.2 |
| Lifetime (days) | 0.55 | 0.57 | 0.62 | 0.03-1.59 |
| Surf concentration (μg/kg) | 12.2 | 18.9 | 16.5 |  |
| Dry deposition (Tg/yr) | 353.6 | 547.9 | 460.9 |  |
| Sedimentation (Tg/yr) | 2049.0 | 3064.3 | 2458.2 |  |
| Kdry (days$^{-1}$) | 1.37 | 1.32 | 1.17 | 0.06-2.94 |
| Wet deposition (Tg/yr) | 278.9 | 417.0 | 354.7 |  |
| SV deposition (Tg/yr) | 505.1 | 771.1 | 746.1 |  |
| Kwet (days$^{-1}$) | 0.45 | 0.43 | 0.44 | 0.11-2.45 |
| SSAOD$_{550nm}$ | 0.0206 | 0.0318 | 0.0269 | 0.003-0.067 |

**Table 2**. Sea salt mass extinction efficient (MEE) for PALMS and GEOS and the ratio of
MEEs between GEOS and PALMS in three vertical layers and in the whole atmosphere
at RH 45%

|  | PALMS (m2/kg) | GEOS (m2/kg) | R(GEOS/PALMS) % |
|---|---|---|---|
| 0 – 1.5 KM | 2636.87 | 1618.09 | 61.4 |
| 1.5 – 6 KM | 2089.97 | 1671.61 | 80.0 |
| >6 KM | 1891.07 | 1786.24 | 94.5 |
| all | 2203.67 | 1679.36 | 76.2 |




**Figure Captions**

**Figure 1**. AToM1 (top) and AToM2 (bottom) flight track sorted out for each flight day.

**Figure 2a**. Scattering plot of sea salt between GEOS and PALMS (magenta) and between GEOS and SAGA (blue) in ATom1 (symbol +) and ATom2 (symbol ◇) for all flight measurements within 1.5 km atmospheric thickness above ocean surface. The SAGA samples are filtered out when dust signal is significant. The GEOS sea salt shown here are cut at 3 μm in dry diameters. Both GEOS and PALMS data are then sampled using SAGA measurement time frequency.

**Figure 2b**. Similar to Figure 2a with the samples contaminated by clouds are further excluded using CAPS cloud indicator.

**Figure 3**. Sea salt ($Dp < 3$ μm) vertical profiles from GEOS simulation and PALMS measurement along ATom1 and 2 flight tracks in 5 latitudinal bands over Pacific and Atlantic oceans. The latitudinal bands are marked by dot grey lines in Figure 1.

**Figure 4**. Total aerosol AOD in 201608 (left column) and 201702 (right column) from MODIS (top) and GEOS (middle). The bottom panel shows the mass fraction of sea salt relative to the total aerosol simulated by GEOS.

**Figure 5**. Total AOD measured by MAN cruise occurred during 201607 to 201706 (5a) and simulated by GEOS but sampled with MAN measurement (5b). 5c shows total AOD





scattering plot between MAN and GEOS and the purple color is for the data over
Southern Ocean shown inside the boxes in Figure 5b.

**Figure 6**. Percentage distribution of sea salt mass over the first three bins normalized to
the total sea salt with particle wet diameter up to ~5 μm at RH 45%. The normalized SS
mass weighting distribution is sorted over three vertical layers and for ATom1 and
ATom2, respectively.

**Figure 7**. Atmospheric RH vertical profiles from GEOS simulation and ATom
measurement along ATom1 and 2 flight tracks in 5 latitudinal bands over Pacific and
Atlantic oceans.











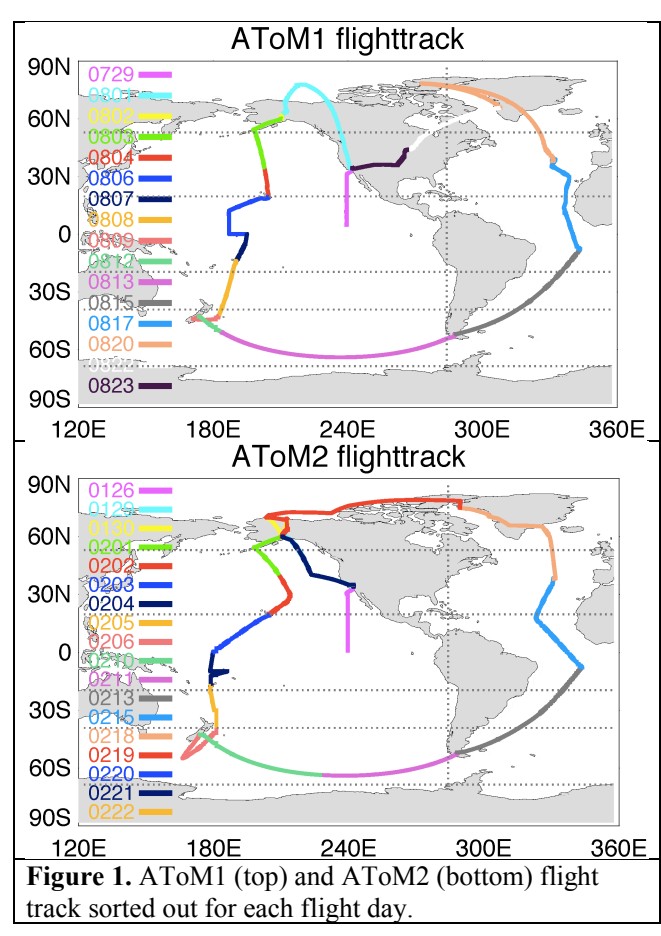

**Figure 1.** AToM1 (top) and AToM2 (bottom) flight track sorted out for each flight day.






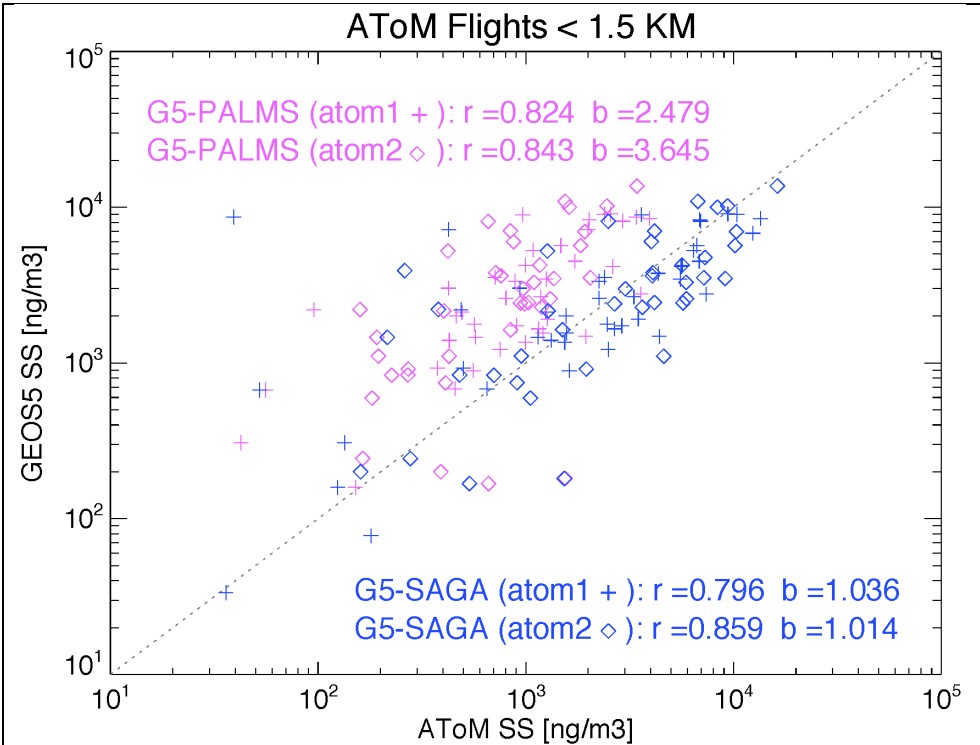

**Figure 2a**. Scattering plot of sea salt between GEOS5 and PALMS (magenta) and between GEOS5 and SAGA (blue) in ATom1 (symbol +) and ATom2 (symbol ◇ ) for all flight measurements within 1.5 km atmospheric thickness above ocean surface. The SAGA samples are filtered out when dust signal is significant. The GEOS5 sea salt shown here are cut at 3 μm in dry diameters. Both GEOS5 and PALMS data are then sampled using SAGA measurement time frequency.






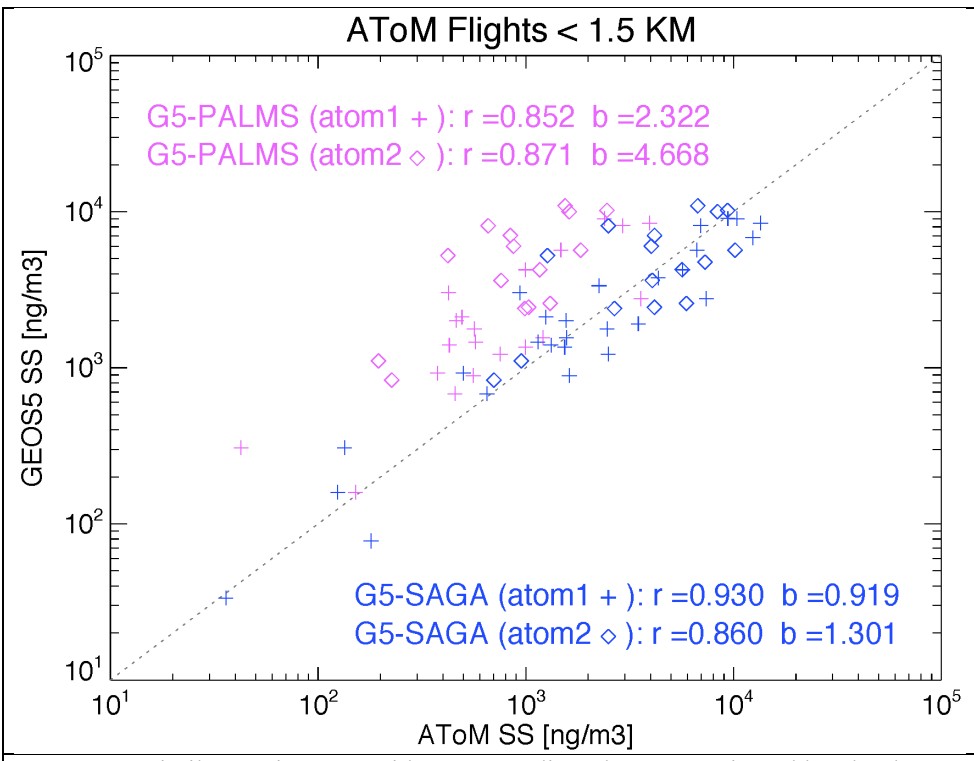

**Figure 2b**. Similar to Figure 2a with SAGA sodium data contaminated by clouds are further excluded using CAPS cloud indicator.





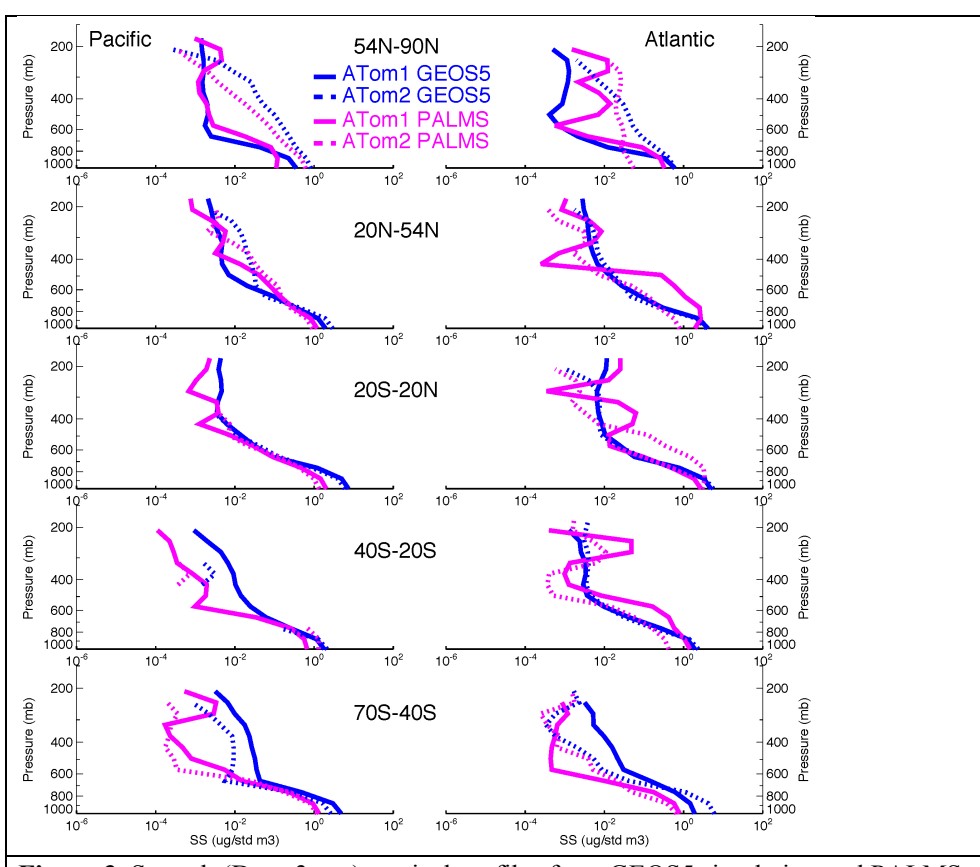

**Figure 3**. Sea salt (Dp < 3 μm) vertical profiles from GEOS5 simulation and PALMS measurement along ATom1 and 2 flight tracks in 5 latitudinal bands over Pacific and Atlantic oceans. The latitudinal bands are marked by dot grey lines in Figure 1.






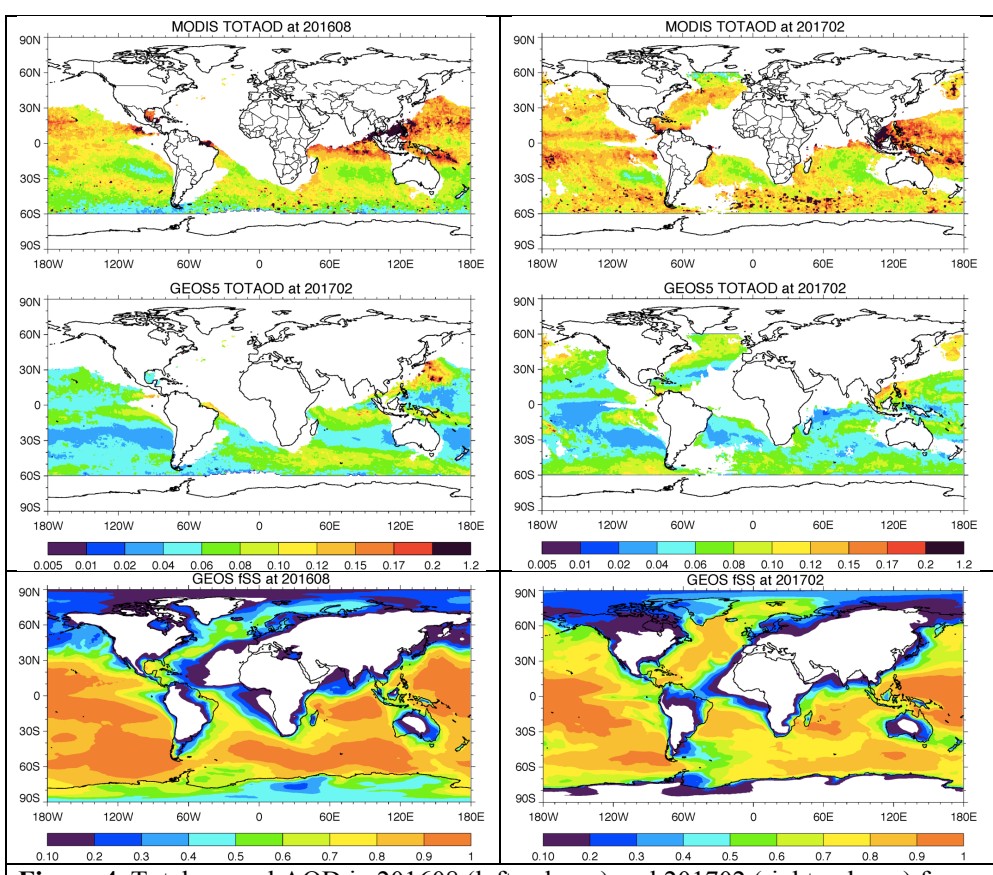

**Figure 4**. Total aerosol AOD in 201608 (left column) and 201702 (right column) from MODIS (top) and GEOS5 (middle) over oceans where fraction of sea salt mass simulated by GEOS (bottom panel) is larger than 0.7.




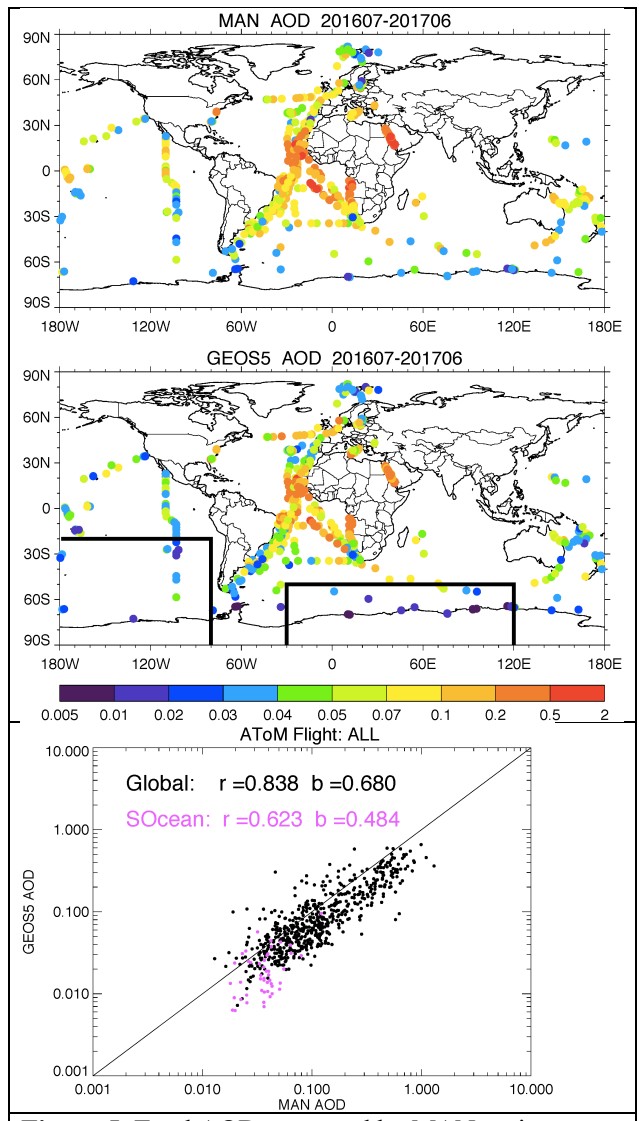

**Figure 5**. Total AOD measured by MAN cruise occurred during 201607 to 201706 (5a) and simulated by GEOS5 but sampled with MAN measurement (5b). 5c shows total AOD scattering plot between MAN and GEOS and the purple color is for the data over Southern Ocean shown inside the boxes in Figure 5b.






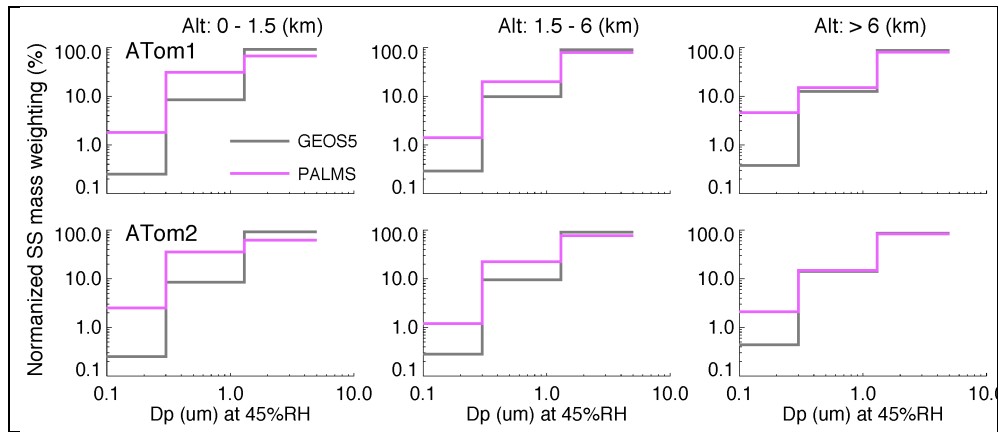

**Figure 6**. Percentage distribution of sea salt mass over the first three bins normalized to the total sea salt with particle wet diameter up to ~5 μm at RH 45%. The normalized SS mass weighting distribution is sorted over three vertical layers and for ATom1 and ATom2, respectively.


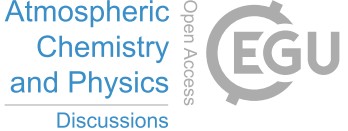

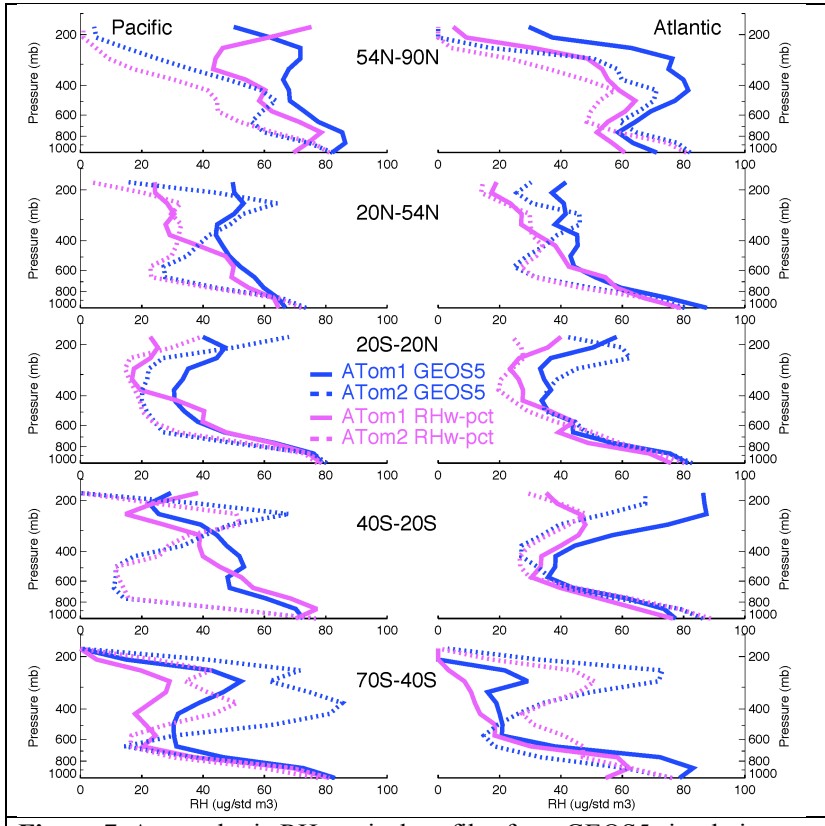

**Figure 7**. Atmospheric RH vertical profiles from GEOS5 simulation and ATom measurement along ATom1 and 2 flight tracks in 5 latitudinal bands over Pacific and Atlantic oceans.
