# Peer review of "Observationally constrained analysis of sea salt aerosol in the"

_Atmospheric Chemistry and Physics, 2019_

## Referee Comment (RC1) · Anonymous Referee #1 · 6 Mar 2019

Review of Observationally constrained analysis of sea salt aerosol in the marine atmosphere by Bian et al.

The manuscript presents valuable inter-comparison between modelled sea spray mass concentration/AOD and extensive in situ measurements. The latter is the most valuable component of this manuscript as vertical distributions of sea spray are indeed not commonly available on the large geographical scale. These measurements provide very good basis for the validation of the model, however, they were not used to their full potential in this manuscript as the appropriate sea spray source function was not provided. The main conclusion that AOD cannot be reproduced by the current model, due to wrong sea spray source function (SSSF) size distribution, is somehow disappointing without providing the appropriate one.

[Figure]

Major comments

In addition to the point raised above, the appropriate comparison of the three SSSF mentioned here is not presented either; There is no discussion or results in the main text, just some numbers in the supplementary, from which it seems that Emi3 results in a higher bias than other schemes. So it is not exactly clear why it was deemed the best here? Manuscript would really benefit from more elaborate discussion on the scheme comparison as well as on how model results compare to AOD measurements using Emi1 and Emi2 schemes? Results should have short description in the main text and only then reference to supplementary (say at lines 117-119);

Introduction section is pretty much biased on USA references, e.g. Quinn and Bates, 2013 is neither the primary nor the main study showing OM in the sea spray; also all other references are mainly from USA scientists, while there are many sea spray papers from European community that were not even mentioned here; For example, extensive SSSF overview paper by (de Leeuw et al., 2011) is missed. Lines 274-275: requires more information and discussion. Is this 0.03 bias comparable with the overestimation here? If not, what percentage is due to bias and what is due to other reasons; Lines 302-304: I understand that the reference is to mass size distribution here, but radiative effects and cloud formation depend more on the number distribution, not mass. Be clear which distribution you refer to and be specific with the effects; Or Lines 313-314, cloud formation is related to size and number not mass; Conclusion on sea water salinity is not convincing globally (lines 400-403), what is salinity variation in global oceans? It might be important locally or regionally close to less saline seas, but not globally; Similarly with the Polar Regions (lines 403-407), indicate how sea ice is relevant to this global study? Is there a higher discrepancy over Polar Regions, if so state that and show the importance? Elaborate on the conclusion sentence in supplementary 'Furthermore, the three emission algorithms discussed in supplementary section show that the uncertainty among the model simulations is generally less than the difference between model and measurement'. First, algorithms do not show anything,

comparison, maybe, second, does this sentence mean that the discrepancy between model and measurements is larger than the model result variation between different SSSF? Clarify. Authors claim that ' Model sensitivity experiments indicated that the simulated sea salt is better correlated with measurements when the sea salt emission is calculated based on the friction velocity and with consideration of sea surface temperature dependence than that parameterized with the 10-m winds' but these results are not properly discussed or presented in the text. Supplementary figures and tables also do not clearly prove that Emi3 is better than other schemes. Correlation might have improved, but the bias got worse. Can you base the conclusion on correlation only?

Specific comments:

Line 161: provide the correlation coefficient; Line 57-58: Dall et al., 2017 reference is not in the list; Quinn et al., 2017 paper says that sea spray is not important for cloud formation, so the reference is not appropriate here Fig. 2: add '3' to superscript in both axis names; Do three significant number have meaning in the correlation coefficient and slopes (are they really so precise?); Fig.2 and lines 188-189: R square is usually presented for model-measurement comparisons, have either R2 or both Lines 245-246: 90% in the mass, not number, provide reference; Lines 281-283: sentence needs rewriting, simulation occurred in July or measurements over this period were compared? Conclusion cannot be obtained in Fig.5. Fig 5 indicates. . .? Lines 300-301: specify what do you mean by 'small particles are more optically efficient' do they scatter better or worse? It is commonly accepted that large particles scatter better; Also, refer to size ranges when talking about small or large particles (here and everywhere in the manuscript) ; E.G line 312: what is small here; Line 318: efficiency? Line 370: with which SSSF the agreement between model and measurements is remarkable? Line 406: Dall et al. 2017 reference is not in the reference list; Table 1: Emi1, Emi2,. . . are not described in the text or table caption; Supplementary Line 40: Emi3 is improved for total mass not size distribution Supplementary Lines 70-71: what do you

mean by shifts? Supplementary Line 75: higher than what? Supplementary Line 82: improvement from 0.5 to 0.54 might be perceived as marginal, no? Why there is such big difference in Atom1 and Atom2 agreements, correlations?

de Leeuw, G., Andreas, E. L., Anguelova, M. D., Fairall, C. W., Lewis, E. R., O'Dowd, C., Schulz, M., and Schwartz, S. E.: Production flux of sea spray aerosol, Rev Geophys, 49, RG2001, doi:10.1029/2010rg000349, 2011.

---

## Referee Comment (RC2) · Anonymous Referee #2 · 24 Mar 2019

General comment:

This manuscript examines the vertical profile of sea salt aerosol concentrations obtained during the NASA Atom campaign, and evaluate the model's capability in reproducing the observations. The Atom observations offer unique vertical distributions of sea salt aerosols over the ocean, and thus provide some critical insight on the source function of sea salt aerosols. In this work, they chose a source function based on the surface friction velocity and sea surface temperature, and found that the model overestimates the observed sea salt aerosol mass concentrations, but underestimates the AOD over the sea salt dominated area. They suggest that it can be due to the discrepancy in modeled size distribution or relative humidity, pointing the necessity for further investigation to improve the sea salt parameterization. Overall, this work provides in-

sightful information on improving parameterization of sea salt aerosols, and I support the publication of this work in ACP if they can address the following specific comments .

Specific comment:

Line 114: How well is the surface friction velocity being represented in the model? For example, what is the range of error when compared to observations?

Line 161: How is the cut-off diameter of SAGA measurement? How does that compared to PALMS? Figure 2ab shows that the modeled SS seems to be underestimated when compared to the SAGA data, and overestimated while compared to the PALMS? Is it potentially due to the different cut-off diameter? What are the measurement uncertainties of SS in PALMS and SAGA?

Line 211: Although the sea salt between two instruments shows high correlations, is it possible that one of the measurements is consistently higher than the other one?

Line229: What is the vertical resolution of PALMS, and how does that compared to GEOS5? It seems that the model had a hard time catching some of the features in the higher troposphere.

Line 269: Just curious, what is the most abundant aerosol over the Arctic Ocean, as sea salt only contribute to 10-50% as shown in Figure 4 bottom?

Line 270: What is the cut-off diameter for the sea salt aerosols in the modeled AOD?

Line 276: Is the underestimate of AOD consistent around the globe? Or certain latitudes/SSTs have relatively smaller underestimates?

Which factor(s) do you think is/are most critical for improving the sea salt parameterization?

What measurement would you suggest to improve the sea salt parameterization?

Table 1ab: Please write out the words or explain in the captions the abbreviation (such as SV deposition).

Figure 1ab: Please explain in the figure captions that what is r(correlation?) and b (bias?).

Figure 3. Please provide the vertical metric in height (km or m) if possible.

Figure 4. Please explain what is fss in the caption.

Comment for Supplement

Line 55: Could you please provides some details on how 2.41 is derived here (or the related reference)?

Line 65: Do you mean the correction factor, T(SST), ranges from 0.0 to 7 here? I tried to calculate it, and it shows that at 36degreeC, the correction factor is 10.63. Also, at -0.1 degree C, it is 0.36? Is this due to rounding? Please double check. Also, please provide a plot of T(SST) versus SST, if possible. And please provide details on how these correction factors are derived (or the related reference).
* * *

---

## Author Comment (AC1) · 21 May 2019

Review of Observationally constrained analysis of sea salt aerosol in the marine atmosphere by Bian et al.

Reviewer: The manuscript presents valuable inter-comparison between modelled sea spray mass concentration/AOD and extensive in situ measurements. The latter is the most valuable component of this manuscript as vertical distributions of sea spray are indeed not commonly available on the large geographical scale. These measurements provide very good basis for the validation of the model, however, they were not used to their full potential in this manuscript as the appropriate sea spray source function

was not provided. The main conclusion that AOD cannot be reproduced by the current model, due to wrong sea spray source function (SSSF) size distribution, is somehow disappointing without providing the appropriate one.

Answer: We thank the reviewer for the insightful comments. We have carefully accounted for the reviewer's comments and suggestions and our point-to-point response is given below.

We appreciate the reviewer's suggestion in the pursuit of establishing a new sea salt source function through our work. However, we carefully examined our currently available observation and model data and believe that the task asked by the reviewer is beyond the scope that our data can support. To achieve that goal, we need experiments that are designed specifically to measure size-resolved sea salt fluxes near air-sea interface. ATom experiment is not designed to derive a sea salt source function at a convincing precision. ATom aircraft measurement is far away from sea surface. The difference between model and measurement is not entirely attributed to sea salt emission. Any uncertainty in removal processes (e.g. wet deposition, dry deposition, and sedimentation) and dynamic transport processes contribute to the difference as well. Furthermore, the size-dependent sedimentation may reshape the sea salt size distribution away from its source regions. Based on the discrepancy between the model-calculated and ATom-observed size distributions, we can only suggest that a modification of emitted sea salt size distribution might be helpful to reduce the discrepancy.

Major comments Reviewer: In addition to the point raised above, the appropriate comparison of the three SSSF mentioned here is not presented either; There is no discussion or results in the main text, just some numbers in the supplementary, from which it seems that Emi3 results in a higher bias than other schemes. So it is not exactly clear why it was deemed the best here? Manuscript would really benefit from more elaborate discussion on the scheme comparison as well as on how model results compare to AOD measurements using Emi1 and Emi2 schemes? Results should have short

description in the main text and only then reference to supplementary (say at lines 117-119);

Answer: In the revision, we have decided to remove the supplementary material for the discussion of sea salt emission algorithms since it is not our main focus of this study.

Each experiment is designed for its specific purpose. ATom aims to provide an unprecedented suite of measurements over global remote oceans, including vertical and seasonal information of aerosol, cloud, meteorological fields. As pointed out by the reviewer, the vertical distributions of sea spray are indeed not commonly available on the large geographical scale. Combining the ATom measurements with other available satellite and ground measurements, we can evaluate our model performance on a broader scale to find out the deficiencies of the model simulation and their potential causes. In this sense, focusing on the small differences between the results using Gong 2003 and the modified ones in GEOS (as shown in the previous Supplement) would provide little help to resolve the differences.

Reviewer: Introduction section is pretty much biased on USA references, e.g. Quinn and Bates, 2013 is neither the primary nor the main study showing OM in the sea spray; also all other references are mainly from USA scientists, while there are many sea spray papers from European community that were not even mentioned here; For example, extensive SSSF overview paper by (de Leeuw et al., 2011) is missed.

Answer: We added the following reference on sea salt study by European scientists. See line 47 and references. de Leeuw, G., Andreas, E. L., Anguelova, M. D., Fairall, C.W., Lewis, E. R., O'Dowd, C., Schulz, M., and Schwartz, S. E.: Production flux of sea spray aerosol, Rev Geophys, 49, RG2001, doi:10.1029/2010rg000349, 2011.

Reviewer: Lines 274-275: requires more information and discussion. Is this 0.03 bias comparable with the overestimation here? If not, what percentage is due to bias and what is due to other reasons;

Answer: To answer the reviewer's question, we calculated the difference of model AOD and MODIS AOD over oceans where the fraction of sea salt AOD is higher than 0.6. Overall, MODIS AOD is larger than model AOD by 0.043 in August 2016 and 0.062 in February 2017. Both differences are larger than the potential positive bias of MODIS AOD, up to 0.03, over oceans. However, it is hard for us to figure out the percentage contributions asked by the reviewer. The reason is that the work of Levy et al. (2013) gave only statistic value of MODIS AOD bias without the information of geophysical location. We added the new calculation and corresponding discussion in section 4.3 lines 284-290.

Reviewer: Lines 302-304: I understand that the reference is to mass size distribution here, but radiative effects and cloud formation depend more on the number distribution, not mass. Be clear which distribution you refer to and be specific with the effects; Or Lines 313- 314, cloud formation is related to size and number not mass;

Answer: We agree with the reviewer that radiative effects and cloud formation depend on the number of aerosol particles. This is why we studied the sea salt size distribution and emphasized the importance of the sea salt aerosols in the fine mode size in the paper. In lines 314-315, we changed the sentence to "Aerosol size also modulates the transport and removal processes. In lines 324-328, we emphasized the role of sea salt number particles by changing the sentence to "The particle sizes here are limited to be less than 3 micrometer in dry diameter due to the size cut of the PALMS inlet. Particles in this range are most important in light extinction and cloud formation with many more sea salt particles in fine mode than in coarse mode on a per unit mass basis.".

Reviewer: Conclusion on sea water salinity is not convincing globally (lines 400-403), what is salinity variation in global oceans? It might be important locally or regionally close to less saline seas, but not globally;

Answer: Yes. Salinity may not be an important factor in sea salt emission on the global scale because it is relatively uniform across the world oceans. But regionally it may

be important as discussed by Grythe et al., (2014). Our model does not account for the salinity impact at all so that the model sea salt results may be low over cold low saline seas, such as the Baltic Sea. We changed the sentences on lines 419-425 in Conclusion to the following. "Consideration of variations in salinity of surface seawater is missing in the GEOS aerosol model. Although salinity may not be an important factor in sea salt emission on the global scale owing to its relatively uniformity across the world oceans, it may be important regionally as discussed by Grythe et al., (2014). Salinity also impacts sea spray aerosol (SSA) size. The dry SSA size distribution shifts towards smaller sizes with lower salinities found in the EMEP intensive campaigns (Barthel et al., 2014)."

Reviewer: Similarly with the Polar Regions (lines 403-407), indicate how sea ice is relevant to this global study? Is there a higher discrepancy over Polar Regions, if so state that and show the importance?

Answer: The study of potential sea salt from sea ice is not directly relevant to the main study of this work. Similar to the discussion for salinity, here we tried to give any other potential improvements on global sea salt simulation based on recent scientific publications and our knowledge.

Reviewer: Elaborate on the conclusion sentence in supplementary 'Furthermore, the three emission algorithms discussed in supplementary section show that the uncertainty among the model simulations is generally less than the difference between model and measurement'. First, algorithms do not show anything, comparison, maybe, second, does this sentence mean that the discrepancy between model and measurements is larger than the model result variation between different SSSF? Clarify. Authors claim that 'Model sensitivity experiments indicated that the simulated sea salt is better correlated with measurements when the sea salt emission is calculated based on the friction velocity and with consideration of sea surface temperature dependence than that parameterized with the 10-m winds' but these results are not properly discussed or presented in the text. Supplementary figures and tables also do not clearly prove

that Emi3 is better than other schemes. Correlation might have improved, but the bias got worse. Can you base the conclusion on correlation only?

Answer: The supplementary material has been removed with the reasons aforementioned.

Specific comments: Reviewer: Line 161: provide the correlation coefficient;

Answer: Line 161: The correlation coefficient between the two instrument measurements was given in section 4.1 lines 220-221.

Reviewer: Line 57-58: Dall et al., 2017 reference is not in the list; Quinn et al., 2017 paper says that sea spray is not important for cloud formation, so the reference is not appropriate here

Answer: Dall et al., 2017 was listed in the reference. We removed Quinn et al., 2017 and added one more recent relevant study of Dall et al., 2018.

Reviewer: Fig. 2: add '3' to superscript in both axis names; Do three significant number have meaning in the correlation coefficient and slopes (are they really so precise?);

Answer: Done. We guess the question here is about the steps we applied for measurement data quality control. Yes, these are necessary steps for our data analyses. Otherwise, the interpreted sea salt measurement will be contaminated by dust-Na+ and clouds. For example, the difference in correlation coefficient and ratio between Figure 2a and 2b is caused by cloud droplets or ice crystals acting like a high-pressure washer to dislodge some of that salt in forward-facing aircraft inlet.

Reviewer: Fig.2 and lines 188-189: R square is usually presented for model-measurement comparisons, have either R2 or both Lines

Answer: Using R or R2 depends on what kind of comparison we investigate. Here we use R to give a point-to-point correlation between the model and measurement data. By the reviewer' suggestion, we added in the text of R2 to estimate the covariance of

the two datasets, see lines 199-200.

Reviewer: 245-246: 90% in the mass, not number, provide reference;

Answer: The sentence (lines 256-257) has been changed to "This is expected because nearly 90% of injected sea salt mass is in coarse mode based on our emission scheme."

Reviewer: Lines 281-283: sentence needs rewriting, simulation occurred in July or measurements over this period were compared? Conclusion cannot be obtained in Fig.5. Fig 5 indicates: : :?

Answer: The sentence (lines296-298 ) has been changed to "MAN measurements from July, 2016 to June 2017 are used in this study. The GEOS model results are sampled at the closest time and location of the ship-based measurements." Figure 5 does show GEOS AOD is significantly lower than MAN AOD over sea salt dominant regions.

Reviewer: Lines 300-301: specify what do you mean by 'small particles are more optically efficient' do they scatter better or worse? It is commonly accepted that large particles scatter better; Also, refer to size ranges when talking about small or large particles (here and everywhere in the manuscript) ; E.G line 312: what is small here;

Answer: It is true that generally the larger a particle is, the more scattering it has. However, traditional aerosol models simulate aerosol masses. Obviously, on a unit mass basis, fine mode sea salt has a larger cross section than that of coarse mode sea salt. To clarify the fine mode sea salt discussed in the paper, a sentence was added in section 2 lines 140-141: "We further classify the first two bins as fine mode and the remaining bins as coarse mode throughout this paper."

Reviewer: Line 318: efficiency?

Answer: Yes. Changed the word to be "efficiency".

Reviewer: Line 370: with which SSSF the agreement between model and measurements is remarkable?

Answer; The discussion was based on our default sea salt emission algorithm. Also refer to the answer on the last two questions of reviewer #2.

Reviewer: Line 406: Dall et al. 2017 reference is not in the reference list;

Answer: See the answer on the question for "line 57-58" above.

Reviewer: Table 1: Emi1, Emi2,: : : are not described in the text or table caption;

Answer: We removed Emi1 and Emi2 according to our discussion above. We also merged Table 1a and 1b to Table 1 and changed text accordingly.

Reviewer: Supplementary Line 40: Emi3 is improved for total mass not size distribution. Supplementary Lines 70-71: what do you mean by shifts? Supplementary Line 75: higher than what? Supplementary Line 82: improvement from 0.5 to 0.54 might be perceived as marginal, no?

Answer: Supplementary material has been removed.

Reviewer: Why there is such big difference in Atom1 and Atom2 agreements, correlations?

Answer: This is not an easy question to answer. Size cut changes and a correction factor may both contribute. Say, if ATom2 MBL was slightly wetter on average than ATom1 then PALMS dry SS size cut would be slightly lower and produce this result. The PALMS team had to apply a correction to the size distribution data in ATom2 (described in Murphy et al., 2019) to make it more consistent with the SAGA measurement. Of course, GEOS sea salt may also have a seasonal bias. We need additional independent measurements to evaluate this issue.

Anonymous Referee #2

General comment: Reviewer: This manuscript examines the vertical profile of sea salt

aerosol concentrations obtained during the NASA Atom campaign, and evaluate the model's capability in reproducing the observations. The Atom observations offer unique vertical distributions of sea salt aerosols over the ocean, and thus provide some critical insight on the source function of sea salt aerosols. In this work, they chose a source function based on the surface friction velocity and sea surface temperature, and found that the model overestimates the observed sea salt aerosol mass concentrations, but underestimates the AOD over the sea salt dominated area. They suggest that it can be due to the discrepancy in modeled size distribution or relative humidity, pointing the necessity for further investigation to improve the sea salt parameterization. Overall, this work provides insightful information on improving parameterization of sea salt aerosols, and I support the publication of this work in ACP if they can address the following specific comments.

Answer: We thank the reviewer for the insightful comments. We have carefully accounted for the reviewer's comments and suggestions and our point-to-point response is given below.

Specific comment: Reviewer: Line 114: How well is the surface friction velocity being represented in the model? For example, what is the range of error when compared to observations?

Answer: The ocean surface wind of GEOS Modern-Era Retrospective Analysis for Research and Applications version 2 (MERRA2) is directly assimilated using two satellite observations, Special Sensor Microwave Imager (SSM/I) and Quick Scatterometer (QuikSCAT) (Rienecker et al., 2011, appendix B). We run the GEOS model using "replay mode", which means every 6h the model dynamic state including these surface winds is set to the state of MERRA2. We added a sentence in lines 122-124. "The model's surface winds are constrained by the two satellite observations, Special Sensor Microwave Imager (SSM/I) and Quick Scatterometer (QuikSCAT) (Rienecker et al., 2011)."

Reviewer: Line 161: How is the cut-off diameter of SAGA measurement? How does that compared to PALMS? Figure 2ab shows that the modeled SS seems to be underestimated when compared to the SAGA data, and overestimated while compared to the PALMS? Is it potentially due to the different cut-off diameter? What are the measurement uncertainties of SS in PALMS and SAGA?

Answer: The cut-off diameter of SAGA measurement is roughly the same as PALMS's under the marine boundary environment according to the study of the DC-8 Inlet Characterization Experiment (DICE) (McNaughton et al., 2007). We added this sentence in lines 177-178: "In other words, the cut-off size of the SAGA instrument is also roughly 3 micrometer in dry diameter." According to the instrument PIs, the uncertainty is not straightforward, but the precision uncertainty of PALMS in SS mass in the MBL is ~10% and overall uncertainty is probably about ~30% (Froyd et al., 2019). The precision uncertainty of SAGA is ~30% as well. Please also see our discussion for the instrument uncertainties in lines 210-216.

Froyd, K. D., Murphy, D. M., Brock, C. A., Campuzano-Jost, P., Dibb, J. E., Jimenez, J.-L., Kupc, A., Middlebrook, A. M., Schill, G. P., Thornhill, K. L., Williamson, C. J., Wilson, J. C., and Ziemba, L. D.: A new method to quantify mineral dust and other aerosol species from aircraft platforms using single particle mass spectrometry, Atmos. Meas. Tech. Discuss., https://doi.org/10.5194/amt-2019-165, in review, 2019.

Reviewer: Line 211: Although the sea salt between two instruments shows high correlations, is it possible that one of the measurements is consistently higher than the other one?

Answer: Yes, it is. The SAGA sea salt mass is consistently higher than the PALMS sea salt.

Reviewer: Line229: What is the vertical resolution of PALMS, and how does that compared to GEOS5? It seems that the model had a hard time catching some of the features in the higher troposphere.

Answer: The vertical profile of PALMS is based on 3-min averages, which gives a vertical resolution of 2.3 km. Statistical noise becomes large at low mass concentrations of ~1-10 ng/m3 and will contribute to the structure in PALMS SS mass in the upper troposphere. In the meanwhile, the vertical resolution of GEOS could reach ~1 km in the upper troposphere. The missing features there in the model data could also be attributed to the model's vertical and long-range transport.

Reviewer: Line 269: Just curious, what is the most abundant aerosol over the Arctic Ocean, as sea salt only contribute to 10-50% as shown in Figure 4 bottom?

Answer: The most abundant aerosols over the Arctic Ocean are sea salt (10-50%), sulfate (up to 40%), dust (up to 30%) and organic carbon (up to 20%) based on the GEOS results.

Reviewer: Line 270: What is the cut-off diameter for the sea salt aerosols in the modeled AOD?

Answer: The cut-off diameter for the sea salt aerosols in the modeled AOD is 20 micrometer. Please refer to the model description in section 2 lines 136-137 for details.

Reviewer: Line 276: Is the underestimate of AOD consistent around the globe? Or certain latitudes/SSTs have relatively smaller underestimates?

Answer: No. The AOD underestimation occurred primarily over ocean regions. In land anthropogenic and dusty pollution areas, the model sometimes overestimates AOD. We did the model and ATom comparison over five latitudinal bands, as shown in Figure 3, and we did not find an obvious latitudinal dependence in the model performance.

Reviewer: Which factor(s) do you think is/are most critical for improving the sea salt parameterization?

Answer: Improvement of sea salt size distribution, particularly the ultrafine particles, is suggested based on our study. Current GEOS model sea salt emission parametrization generally gives a low-bound cut-off diameter at around 100 nm in dry diameter. This

seems not sufficient, and we suggest to extend it down to 10 nm. Particles smaller than 80 nm in diameter can effectively become CCN through heterogeneous growth and coagulation with other sub-80 nm particles [Clarke et al., 2006], although generally a minimum dry diameter of 80 nm is considered for cloud activation [Pierce et al., 2006].

Clarke, A. D., Owens, S. R. & Zhou, J. 2006 An ultrafine sea-salt flux from breaking waves: implications for cloud condensation nuclei in the remote marine atmosphere. J. Geophys. Res. 111, D06202. (doi:10.1029/2005JD006565)

Pierce, J. R. and Adams, P. J.: Global evaluation of CCN formation by direct emission of sea salt and growth of ultrafine sea salt, J. Geophys. Res., 111, D06203, doi:10.1029/2005JD006186, 2006.

Reviewer: What measurement would you suggest to improve the sea salt parameterization?

Answer: To improve the sea salt parameterization, we need to put more effort on the measurements of size-resolved sea salt flux at various ocean surfaces, such as oceans with different latitudes, seasons, winds, temperatures, salinities, and marine ecosystems. We need to pay particular attention to ultrafine sea salt.

Reviewer: Table 1ab: Please write out the words or explain in the captions the abbreviation (such as SV deposition).

Answer: Done.

Reviewer: Figure 1ab: Please explain in the figure captions that what is r(correlation?) and b (bias?).

Answer: Done for Figure 2ab. Here, the statistical parameter r is the correlation coefficient and b is the ratio of SS(GEOS) to SS(ATom).

Reviewer: Figure 3. Please provide the vertical metric in height (km or m) if possible.

Answer: Done.

Reviewer: Figure 4. Please explain what is fss in the caption.

Answer: Done. We changed fSS to fSSAOD, which is the fraction of sea salt AOD versus total aerosol AOD.

Comment for Supplement Reviewer: Line 55: Could you please provides some details on how 2.41 is derived here (or the related reference)?

Answer: The functional form of the wind- and SST-dependent terms were developed and used in the MERRA2 meteorology and aerosol reanalysis (Darmenov et al, 2013; Randles et al., 2013). Examination of the wind term based on Gong's parameterization was prompted by the presence of high/low bias in sea salt aerosol optical depth (AOD) in high/low latitudes in the GEOS model. To address this bias we analyzed the relationship between sea salt AOD and friction velocity, and concluded that the power factor of 3.41 is too high and should be lowered by about unity (or 2.41). With that change the sea salt emissions and sea salt AOD in GEOS became more uniform and with less pronounced zonal gradient. We would like to point out that the power factor of 2.41 is well within the range of values reported by other studies (see for example compilation of 10-meter wind and friction velocity parameterizations by Anguelova et al. (2006) and more recently by Brumer et al. (2017)). Similarly to Jaegle et al. (2011), we examined the remaining differences between the model and satellite AODs (when sea salt had significant contribution to the total AOD), and attributed these to the effects of SST on sea salt emissions by parameterizing the ratio of observed to modeled AOD as a function of SST. The SST used at the time in GEOS was from the Reynolds dataset.

This work is not attempting to develop or modify the GEOS sea salt emission. Rather we intend to suggest a direction in improving emitted sea salt size distribution that might be helpful to reduce the discrepancy between the model-calculated and ATom-observed size distributions. Please also refer to our answers to the major comment of reviewer #1

Anguelova, M. D., and F. Webster (2006), Whitecap coverage from satellite measurements: A first step toward modeling the variability of oceanic whitecaps, J. Geophys. Res., 111, C03017, doi:10.1029/2005JC003158.

Brumer, S.E., C.J. Zappa, I.M. Brooks, H. Tamura, S.M. Brown, B.W. Blomquist, C.W. Fairall, and A. Cifuentes-Lorenzen, 2017: Whitecap Coverage Dependence on Wind and Wave Statistics as Observed during SO GasEx and HiWinGS. J. Phys. Oceanogr., 47, 2211–2235, https://doi.org/10.1175/JPO-D-17-0005.1

Darmenov, A., da Silva, A., Liu, X. and Colarco, P. R., (2013), Data-driven aerosol development in the GEOS-5 modeling and data assimilation system, Abstract A43D-0305 presented at 2013 Fall Meeting, AGU, San Francisco, Calif., 9-13 Dec.

Jaeglé, L., Quinn, P. K., Bates, T. S., Alexander, B., and Lin, J.-T.: Global distribution of sea salt aerosols: new constraints from in situ and remote sensing observations, Atmos. Chem. Phys., 11, 3137-3157, https://doi.org/10.5194/acp-11-3137-2011, 2011

Randles, C.A., A.M. da Silva, V. Buchard, P.R. Colarco, A. Darmenov, R. Govindaraju, A. Smirnov, B. Holben, R. Ferrare, J. Hair, Y. Shinozuka, and C.J. Flynn, 2017: The MERRA-2 Aerosol Reanalysis, 1980 Onward. Part I: System Description and Data Assimilation Evaluation. J. Climate, 30, 6823–6850, https://doi.org/10.1175/JCLI-D-16-0609.1

Reviewer: Line 65: Do you mean the correction factor, T(SST), ranges from 0.0 to 7 here? I tried to calculate it, and it shows that at 36degreeC, the correction factor is 10.63. Also, at -0.1 degree C, it is 0.36? Is this due to rounding? Please double check. Also, please provide a plot of T(SST) versus SST, if possible. And please provide details on how these correction factors are derived (or the related reference).

Answer: Your calculation is right. The T(SST) will start from 0.4 when SST is close to frozen point. Since T(SST) is confined to be less than 7, the corresponding up-bound SST should be around 34.6. In our model calculation, we run SST from 0 up to 36.0 and reset T(SST) to be 7 when it is larger than 7. The figure of T(SST) versus SST is

provided here. Please refer the answer to "Line 55" for the derivation of temperature correction.

Figure shows the relationship between T(SST) and SST.
[Figure]

[Figure]

Figure shows the relationship between T(SST) and SST.

**Fig. 1.**

---

## Author Response (AR2)

Observationally constrained analysis of sea salt aerosol in the marine atmosphere

**Co-Editor Decision: Publish subject to technical corrections** (16 Jul 2019) by Maria Cristina Facchini
Comments to the Author:
Please follow the suggestions of technical corrections indicated by Referee #1

Figure 2: include only 2 significant numbers for the r (e.g. 0.82 instead of 0.824);
*Answer: Done.*

Lines 220-221: add info on instrument comparison other than correlation (e.g. if the SAGA sea salt mass is consistently higher than the PALMS sea salt as refereed in the response to the reviewer, provide this info here and add quantification: 30% higher, 2 times, how much?)
*Answer: The sentence has been changed as: "Comparing sea salt between the two instruments directly shows a high correlation (0.81 in ATom1 and 0.94 in ATom2), while sea salt mass of PALMS is only 36% (ATom1) and 24% (ATom2) of that in SAGA (also see Murphy et al., 2019)."*

Dall et al. is still not in the reference list, did you mean Dall'Osto? Use the full name, not half
*Answer: Change Dall to Dall'Osto in line 56 and 428.*